# Comparison of Bone Tissue Trace Element Content in the Different Radiological Stages of Hip Osteoarthritis

**DOI:** 10.3390/ijerph18063260

**Published:** 2021-03-22

**Authors:** Mikołaj Dąbrowski, Anetta Zioła-Frankowska, Marcin Frankowski, Jacek Kaczmarczyk, Łukasz Kubaszewski

**Affiliations:** 1Adult Spine Orthopaedics Department, Poznan University of Medical Sciences, 61-545 Poznan, Poland; pismiennictwo1@gmail.com; 2Department of Analytical Chemistry, Faculty of Chemistry, Adam Mickiewicz University in Poznan, 61-614 Poznan, Poland; anettazf@amu.edu.pl; 3Department of Analytical and Environmental Chemistry, Faculty of Chemistry, Adam Mickiewicz University in Poznan, 61-614 Poznan, Poland; marcin.frankowski@amu.edu.pl; 4Poznan LAB, Institute of Practical Medicine, 62-081 Przezmierowo, Poland; drkaczmarczyk@tlen.pl

**Keywords:** trace elements, metalloproteinase, femoral bone, hip osteoarthritis stages, grades

## Abstract

Bone metabolism and the trace element content associated with it change at each stage of degenerative disease. The aim of this study was to find out about the role of the analyzed elements in different stages of hip osteoarthritis. Elements associated with oxidative and enzymatic processes were analyzed depending on the changes in the radiological images of the hip joint. Element content analysis was performed by the inductively coupled plasma mass spectrometry analytical technique. The femoral head in severely osteoarthritic hips (KL3–4) compared to mild grade osteoarthritis (KL2) had a greater content of Cu (median 1.04 vs. 0.04), Sr (median 38.71 vs. 29.59), and Zn (median 75.12 vs. 63.21). There were no significant differences in the content of Mo, Cr, and Fe in the femoral head and neck between the groups. The Cu/Fe correlation was negative in the KL2 group (−0.47) and positive in the KL3–4 groups (0.45). Changes in the content and correlation of trace elements in the hip joint explain the changes in metabolism dependent on the severity of degenerative changes.

## 1. Introduction

Osteoarthritis (OA) is a common disease that mainly involves cartilage destruction, synovial inflammation, osteophyte formation, and subchondral bone sclerosis [1,2]. The content of enzymatic and oxidative elements and their correlations in bone tissue have so far been analyzed in OA of the hip joints [3,4], knee joints [5,6], and intervertebral disc [7,8].

Enzymatic elements (Ca, Zn, Cu, and Mo) are cofactors of enzymes, e.g., metalloproteinases (MMPs), involved in the pathogenesis of degenerative joint lesions. Interleukin-1 beta (IL-1ß) and tumor necrosis factor alpha (TNF-α) stimulate the production of matrix-degrading enzymes such as disintegrin and thrombospondin motif metalloproteinases (ADAMTS) produced by macrophages, fibroblasts, and chondrocytes [1]. Degenerative changes depend on oxidative processes, which are necessary enzyme cofactors, such as superoxide dismutase (Cu and Zn), catalase (Cu and Fe), and various types of glutathione peroxidases (Se) [9].

The influence of the severity of degenerative changes on the content of metals and their correlation in femoral bone has not been investigated in humans. The aim of this study was to present the role of the analyzed elements in the pathogenesis of OA. This study will help indirectly show metabolic changes taking place in the bone in response to the progressive degenerative process.

## 2. Materials and Methods

### 2.1. Patients

The sample consisted of 58 patients who underwent total hip replacement (THR). All patients lived in the Greater Poland region of Poland, a region that has no major industry concentration [4]. Table 1 shows the characteristics of the patients enrolled in the study.

The inclusion criteria were age >60 years, primary OA, no exposure to environment pollution of air, water, or soil, and no contact with chemicals (i.e., fertilizers, plant protection arable, heavy metals, paints, adhesives, plaster, tannery articles, photographic chemicals, reagents used in metallurgy, other chemicals, and detergents). The interview of exposure to environmental pollutants was conducted with the use of a questionnaire completed during a hospital stay.

The exclusion criteria were secondary degenerative changes and hip fractures, history of cancer, liver or kidney failure, heart failure of class III or IV according to the New York Heart Association (NYHA), and taking drugs that affect bone metabolism, such as mineral supplements, neuroleptics, chemotherapeutic agents, immunosuppressive drugs, corticosteroids, or antidepressants.

### 2.2. Characterization and Sampling of Femoral Bone Samples and Determination the Elemental Metals

The bone samples were prepared based on the method described by Zioła-Frankowska et al. [10]. Directly after acquisition, the trabecular subchondral bone was separated from the femoral heads under sterile conditions. Samples were cut from the head and neck of the femur. In the case of the femoral neck, samples were collected with a patch section thickness of 1–2 mm, and a 5 mm slice was taken in the shape of a triangle. After preparation, the concentration of selected metals was determined using an ICP-OES spectrometer (VISTA-MPX; VARIAN, Victoria, Mulgrave, Australia).

### 2.3. Radiographic Assessment

All of the subjects included in this study were undergoing bilateral anteroposterior radiography of the hip. Two orthopedists, blinded to the subjects’ clinical symptoms, assessed the radiographs independently using the Kellgren–Lawrence (KL) radiographic system. Stage 2 is described as definite osteophyte formation with possible joint space narrowing; stage 3 as multiple osteophytes, definite joint space narrowing, sclerosis, and possible bony deformity; stage 4 as large osteophytes, marked joint space narrowing, severe sclerosis, and definite bony deformity. A participant was diagnosed with radiographic hip OA if at least one of the hip joints was graded as KL grade 2 or above. Two groups of patients were selected: Mild OA (KL2) and severe OA (KL3–4).

### 2.4. Statistical and Chemometrics Analysis

The analysis used Statistica software (Version 13.0, StatSoft Inc., Tulsa, OK, USA). To compare the impact of various environmental factors on the concentration of analyzed metals in the bone, we used a Mann–Whitney *U*-test (*p* < 0.05). In addition, we determined the Spearman’s rank correlation between the trace elements occurring in the materials and between the different studied metals in different stages of OA. Chemometric analysis was performed to evaluate variables from the independent assumption showing the mutual relationships between the analyzed factors by applying principal component analysis (PCA).

## 3. Results

The study included 22 patients with mild grade OA (KL2) and 36 with severe-grade OA (KL3–4). The mean age and number of women and men did not differ significantly between the groups. Significantly greater pain intensity was demonstrated in patients with a severe radiological stage of degenerative disease (7.8 ± 0.8 vs. 7.2 ± 0.6, *p* = 0.01) (Table 1).

There was a significantly higher content of Cu in the femoral head (median 1.04) in severe-grade OA (KL3–4) compared to the mild stage (KL2) (median 0.04) (*p* = 0.03). A significantly higher Sr content (median 38.71) was found in the femoral head in the KL3–4 group compared to the KL2 group (median 29.59) (*p* = 0.01). The Zn content was higher in patients with severe vs. mild OA—72.21 vs. 63.21 mg·kg^−1^ (*p* = 0.06). There were no significant differences in the contents of Mo, Cr, and Fe in the femoral head and neck between the groups (Table 2).

A significantly higher content of structural elements in the femoral head in severe-grade OA was demonstrated compared to mild-grade OA: Ca, 143.95 vs. 111.96 g·kg^−1^ (*p* = 0.01); Mg, 1.57 vs. 1.23 (*p* < 0.01); P, 66.29 vs. 52.1 (*p* = 0.01), respectively. There were no significant differences in the content of the above elements in the femoral neck between the groups (Table 2).

There was a strong positive Cu/Cr correlation (0.74) in the KL3–4 group compared to the KL2 group (−0.27) in the femoral head, and in the femoral neck, the correlations were 0.21 (KL3–4) and 0.33 (KL2) (Table 3 and Table 4). The Spearman’s coefficient of the Cu/Fe correlation was negative in the KL2 group (−0.47) and positive in the KL3–4 group (0.45). The Cu/Fe correlation in the femoral neck was not significant—0.30 (KL2) and 0.39 (KL3–4). The significant Fe/Cr correlation in the KL2 group was 0.51 (in the femoral head) and 0.64 (in the femoral neck) and 0.25–0.26 in the KL3–4 group. Significant negative correlations were found for Mo/Cu (−0.46) and Mo/Sr (−0.45) in the KL2 group in the femoral head, which were not present in the KL3–4 group (Mo/Cu, 0.16; Mo/Sr, 0) (Table 3). Significant Mo/Cu and Mo/Sr correlations were not present in the femoral neck (Table 4).

PCA in the femoral head showed a difference between the OA hip grades for the first factor for Cu (KL3–4), Zn, and Fe (KL2). The second factor describes the difference between the groups of Zn (KL2) and Cu (KL3–4). In the femoral neck, there was a difference between the OA hip grades for the first factor for Mo (KL3–4) and Zn (KL2), and for second factor for Cu (KL3–4) and Zn (KL2) (Figure 1).

## 4. Discussion

This study compared the concentrations of trace and structural elements in patients with different severities of OA in their hip joints. As expected, patients with radiographic severe-grade OA had greater pain intensity. Analysis of the correlation results, depending on the severity of hip OA, showed differences in the contents of Cu, Zn, and Sr, as well as the structural elements Ca, P, and Mg only in the femoral head. However, there were no significant differences in the content of the other elements in the femoral head and neck. In this study, it was shown that the content of Zn in the femoral head in patients with degenerative changes of the hip increased with their severity, although the difference was at the borderline of statistical significance.

Mahmood et al. showed that serum Zn was significantly lower in OA patients compared to the control group [11]. The role of Zn in the protection of cartilage is to protect against the action of pro-inflammatory cytokines (IL-1β) through the activation of antioxidant genes and participation in the production and conformation of collagen. Increasing the concentration of collagen type X (ColX) in patients with hip OA indicates its role in the pathogenesis of the disease [12,13]. Increased Zn concentrations lead to the activation of catabolic pathways in chondrocytes and an increase in the activity of matrix metalloproteinases [12,14,15]. Increased concentrations of MMP-1, 3, 9, and 13 have been shown in the subchondral bone, cartilage, and synovial membrane secreted from chondrocyte specimens in OA patients. The overexpression of zinc transporter (ZIP8) in chondrocytes leads to increased levels of intracellular Zn and matrix-degrading enzymes of Zn-dependent metalloproteases [1,16]. Zn binds specifically to metallothionein (MT), which regulates metal homeostasis by transferring bound Zn ions to zinc-dependent enzymes and transcription factors, leading to changes in enzyme activity and DNA binding affinity [1]. It has been shown that lower Zn concentrations can produce osteogenic effects [17]. 

In this study, significantly higher Cu concentrations were found in patients with severe hip OA, indicating an important role of Cu in degenerative changes in joints. The Cu concentration in synovial fluid was significantly higher in patients with OA than in healthy subjects [15]. Kubaszewski et al. noticed increased concentrations of Cu in the degenerated intervertebral disc [8]. Previous studies have shown a significant increase in the serum Cu concentration and the Cu/Zn correlation in patients with OA of the knee joint and their duration and severity [11]. The serum Cu/Zn correlation was significantly higher in aging compared to middle-aged adults, and it was significantly increased in elderly patients compared to healthy controls. The Cu/Zn ratio in healthy elderly subjects was due to higher Cu concentrations, while patients had higher Cu and lower Zn serum concentrations. The Cu/Zn ratio is bound by lipid peroxidation products, which suggests a relationship with systemic oxidative stress [18]. Our study did not show that the Cu/Zn correlation was significantly different between the groups of hip OA severity. 

Fe is essential for the hydroxylation of proline and lysine residues in biosynthetic collagen precursors in bone tissue, as well as a cofactor of 25-hydroxycholecalciferol hydroxylase. Intracellular iron accumulation is an important feature of the aging process, especially in post-mitotic tissues, with both hem-Fe and heme biosynthesis known to significantly decrease with age [18]. Osteophytes occur at an early stage of the disease under the influence of TGFβ1 and possibly bone morphogenetic proteins (BMPs)/Smad signaling pathways [19]. It was shown that the concentration of Cu and Fe in the synovial fluid was significantly higher in the group of patients with OA of the knee joint. Similar results were obtained for Fe, where the concentrations in the synovium of patients with OA were slightly higher than normal. The bone Fe content of dogs in the severe OA group was significantly higher than that of non-degenerative bones [20]. 

In our study, we found no differences in the Fe content in the femurs between groups, with major and minor osteophytes (KL2 vs. KL3–4). The occurrence of differences in the concentrations of Cu, Fe, and Zn only in the femoral head and not in the femoral neck may confirm the metabolic involvement of these elements in hip OA.

Our study showed a significantly higher content of Sr in patients with severe OA of the hip. This may be related to the activity of Sr, which could promote collagen synthesis and suppress collagen degradation via the repression of MMP-13 expression [21]. In addition, it has been shown that younger people or those with bone pathologies that increase the rate of remodeling accumulate more Sr than healthy adults [22]. Therefore, a higher strontium concentration in more advanced degenerative disease likely results from subchondral remodeling and is analogous to changes in Ca and P. 

Our study showed a significantly higher content of Mg in the femoral head in patients with severe OA. Benlidayi et al. found that the severe OA group had significantly lower serum Mg levels than the mild OA group and was independent of the inflammation of ESR and CRP [23]. In turn, intra-articular Mg injections reduce OA progression in a rat model. This effect is at least partly explained by the stimulation of cartilage matrix synthesis and the suppression of synovitis [24]. A correlation has also been demonstrated between a diet rich in Mg and a decrease in the intensity of radiographic changes in osteoarthritis of the knee joint [25]. Mg deficiency can affect articular cartilage loss, abnormal bone formation, and tissue inflammation. Mg deficiency causes the onset and progression of osteoarthritis. In addition, Mg nutritional supplementation or local infiltration may be a prophylactic treatment and may slow the progression of osteoarthritis [26]. Higher consumption of Mg in the diet is associated with a significant increase in the cartilage of the knee joint and its volume confirmed by magnetic resonance imaging [27]. 

The limitation of the study is the relatively small number of patients without a control group. We did not have access to healthy femurs (we did not remove the femur in healthy people and we did not have access to dissection materials). Therefore, we compared the results with the data in the literature. We are aware that the content of metals is influenced by many factors, not only enzymatic or oxidative elements, but also environmental factors that are difficult to exclude. Part of the correlation occurring in OA of the hip has been demonstrated by other authors, which may confirm their participation in degenerative changes. Although the examined patients came from a fairly homogeneous environment, we cannot exclude all of the sources of the elements. Moreover, the conclusions of the study regarding metabolic changes in degenerative disease are obviously indirect.

## 5. Conclusions

Our results indirectly confirm that there are differences in the metabolism of femoral bone depending on the severity of OA. We showed significant differences in the content of the five elements Cu, Zn, Sr, Mg, Ca, and P in the femoral head between mild and severe OA. The contents of metals and their correlations in various tissues can therefore be treated as a marker of oxidative stress and metabolic changes. Therefore, the correlations may indirectly explain the oxidative and enzymatic causes of the OA and investigate the mechanisms related to the formation of osteophytes and subchondral sclerosis.

## Figures and Tables

**Figure 1 ijerph-18-03260-f001:**
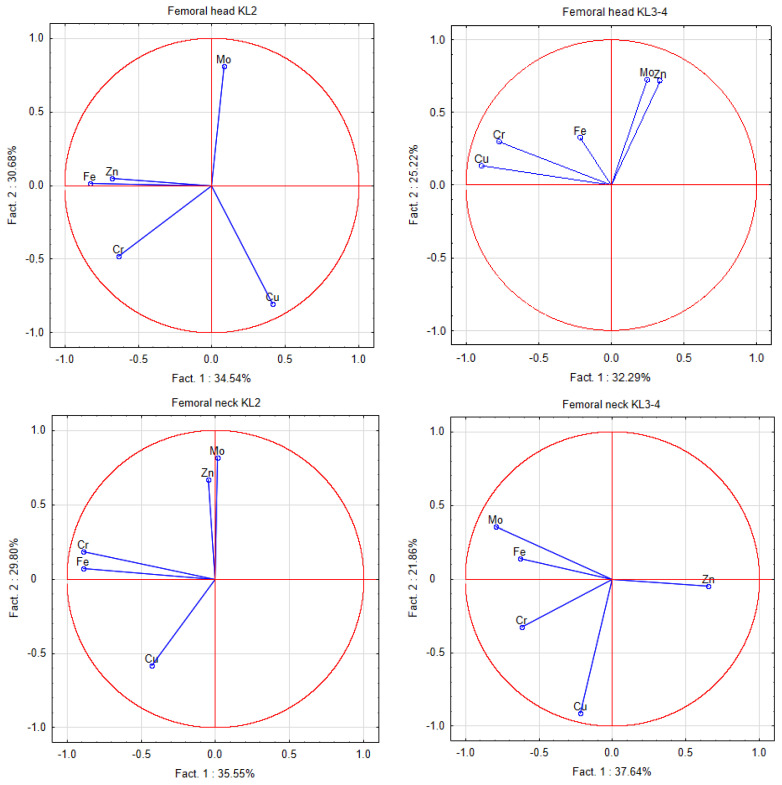
A graphic illustration of principal components analysis of the contents of elements in the femoral head and neck according to the Kellgren–Lawrence scale of hip OA. Projection of the variables on the factor plane of the first two principal components for the mild and severe grades of OA.

**Table 1 ijerph-18-03260-t001:** Information on patients included in the study (*N* = 58).

		AM ± SD/Min. and Max.
Age (years)	All patients	70.6 ± 10.4/64–78
	KL2	72 ± 8.9/64–79
	KL3–4	69.7 ± 11.2/64–77.5
BMI (kg·m^−2^)	All patients	27.8 ± 5.1/24.3–30.5
	KL2	27.1 ± 4.3/25–29.9
	KL3–4	28.3 ± 5.6/24.1–30.8
VAS	All patients	7.6 ± 0.8/7–8
	KL2	7.2 ± 0.6/7–8
	KL3–4	7.8 ± 0.8/7–8
		Number of patients (%)
Sex (Female/Male)	All patients	38 (65.5%)/20 (34.5%)
	KL2	16 (72.7%)/6 (27.3%)
	KL3–4	22 (61.1%)/14 (38.9%)
Place of residence	Village	13 (22.4%)
	City < 10,000	11 (19%)
	City > 10,000	34 (58.6%)
Cigarette smoking	Nonsmoker	48 (82.8%)
	Irregular smoker	4 (6.9%)
	Regular smoker	6 (10.3%)
Alcohol drinking	Nondrinker	30 (51.8%)
	Occasionally	14 (24.1%)
	Often	14 (24.1%)
Comorbidities	Diabetes	11 (19%)
	Arterial hypertension	38 (65.5%)
	Other heart diseases	8 (13.8%)
	Other	14 (24.1%)

VAS, visual analogue scale; BMI, body mass index; AM, arithmetic mean; SD, standard deviation; KL, Kellgren–Lawrence radiographic grading system.

**Table 2 ijerph-18-03260-t002:** Concentrations of enzymatic and oxidation metals (Mo, Cr, Zn, Cu, Fe, Sr in mg·kg^−1^ on dry mass basis) and structural elements (Ca, P, Mg in g·kg^−1^ on dry mass basis) differences between mild and severe grade of OA in femoral head and neck (*N* = 58).

	Femoral Head			Femoral Neck		
Metal	KL2 (*n* = 22)	KL3–4 (*n* = 36)	*p* MW	KL2 (*n* = 22)	KL3–4 (*n* = 36)	*p* MW
	AM ± SD	AM ± SD		AM ± SD	AM ± SD	
	Med. (IQR)	Med. (IQR)		Med. (IQR)	Med. (IQR)	
Mo	0.59 ± 0.56	0.54 ± 0.62	NS	0.8 ± 0.62	0.75 ± 0.82	NS
	0.18 (0.18–1.15)	0.18 (0.18–1.08)		0.71 (0.18–1.38)	0.18 (0.18–1.45)	
Cr	0.91 ± 0.98	1.44 ± 2.17	NS	1.56 ± 1.8	1.56 ± 1.82	NS
	0.42 (0.12–1.68)	0.49 (0.12–1.79)		0.76 (0.24–2.05)	0.85 (0.25–1.73)	
Zn	66.52 ± 16.65	75.12 ± 17.07	0.06	64.86 ± 13.3	68.17 ± 14.34	NS
	63.21 (54.82–75.8)	72.21 (63.89–90.47)		62.7 (55.05–71.86)	68.24 (57.87–75.44)	
Cu	0.62 ± 0.79	1.14 ± 0.96	0.03 *	0.89 ± 0.9	1.12 ± 1.51	NS
	0.04 (0.04–1.37)	1.04 (0.42–1.61)		0.68 (0.04–1.58)	0.8 (0.04–1.31)	
Fe	112.18 ± 78.28	147.72 ± 131.91	NS	152.36 ± 187	129.08 ± 110.45	NS
	89.54 (66.04–118.3)	113.96 (54.82–144.13)		82.07 (47.13–155.29)	107.05 (55.71–164.66)	
Sr	33.74 ± 15.2	49.94 ± 29.65	0.01 *	39.44 ± 17.45	46.45 ± 20.22	NS
	29.59 (23.38–37.97)	38.71 (29.76–59.66)		34.75 (27.18–46.61)	42.81 (29.83–58.01)	
Ca	122.07 ± 36.18	145.24 ± 33.98	0.01 *	149.36 ± 42.98	153.37 ± 41.95	NS
	111.96 (97.2–130.33)	143.95 (115.68–167.91)		135.94 (121.05–171.4)	151.21 (125.96–177.87)	
Mg	1.29 ± 0.44	1.54 ± 0.31	0.00 *	1.49 ± 0.35	1.57 ± 0.34	NS
	1.23 (0.95–1.4)	1.57 (1.31–1.8)		1.44 (1.26–1.75)	1.61 (1.37–1.81)	
P	55.49 ± 16.78	66.48 ± 15.69	0.01 *	67.02 ± 18.41	68.85 ± 19.36	NS
	52.1 (42.99–58.39)	66.29 (52.6–78.89)		62.75 (53.6–81)	68.03 (56.96–77.81)	

MW, Mann–Whitney test; NS, non-significant; IQR, interquartile range; AM, arithmetic mean; SD, standard deviation. * Statistical significance.

**Table 3 ijerph-18-03260-t003:** Spearman’s correlation coefficients for the metals found in the femoral head for mild and severe grades of osteoarthritis (OA).

Femoral Head	Mo	Cr	Zn	Cu	Fe	Sr
	KL3–4					
Mo	x	−0.02	0.27	−0.16	−0.08	0
Cr	−0.18	x	−0.12	0.74 *	0.26	−0.14
Zn	−0.17	0.26	x	−0.07	0.11	0.57 *
Cu	−0.46 *	0.11	−0.27	x	0.45 *	−0.16
Fe	0.05	0.51 *	0.17	−0.47 *	x	−0.02
Sr	−0.45 *	0.30	0.63 *	−0.06	0.26	x
	KL2					

* Statistical significance.

**Table 4 ijerph-18-03260-t004:** Spearman’s correlation coefficients for the metals found in the femoral neck for mild and severe grades of OA.

Femoral Neck	Mo	Cr	Zn	Cu	Fe	Sr
	KL3–4					
Mo	x	0.30	−0.46 *	−0.02	0.20	−0.33 *
Cr	0.11	x	0.03	0.21	0.25	0.00
Zn	0.34	−0.09	x	−0.05	−0.20	0.48 *
Cu	−0.25	0.33	−0.06	x	0.39	−0.35 *
Fe	−0.09	0.64 *	−0.27	0.30	x	−0.23
Sr	0.31	−0.08	0.43 *	−0.02	−0.03	x
	KL2					

* Statistical significance.

## Data Availability

This data can be found at: https://1drv.ms/x/s!AmJVkCPyMge3kADmkGCWfaz6cPkq (accessed on 22 March 2021).

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
