# Peer review of "Comparison of Bone Tissue Trace Element Content in the Different Radiological Stages of Hip Osteoarthritis"

_ijerph, 2021, doi:10.3390/ijerph18063260_

Round 1
Reviewer 1 Report
The current version of the manuscript has improved. There are still some points to fix:
In table 1: report in the legend the meaning of acronyms BMI, VAS, AM and SD.
Omit in the first row the term “Factors”
In table 2: why was in bold the p (0.06) of the difference in Zn levels between mild and severe grade of OA in femoral head? Only p<0.05 should be significant (see also results section).
Report in the legend the meaning of acronyms AM and SD.
In table 3 and 4 report in the legend the meaning of symbol “*” as significant correlations
Author Response
Dear Reviewer,
Thank you for your valuable comments and suggestions for improvements.
P1. In table 1: report in the legend the meaning of acronyms BMI, VAS, AM and SD.
A1. Thanks for your suggestion. A legend has been added to the table.
P2. Omit in the first row the term “Factors”
A2. Thanks for your suggestion. Table description deleted.
P3. In table 2: why was in bold the p (0.06) of the difference in Zn levels between mild and severe grade of OA in femoral head? Only p<0.05 should be significant (see also results section).
A3. Thanks for your suggestion. Bolding removed.
P4. Report in the legend the meaning of acronyms AM and SD.
A4. Thanks for your suggestion. The legend has been completed in the table.
P5. In table 3 and 4 report in the legend the meaning of symbol “*” as significant correlations
A5. Thanks for your suggestion. Legends have been added to the tables.
We would like to thank for the time and effort. We sincerely hope that the revised manuscript could meet your requirement for its publication.
Reviewer 2 Report
The manuscript has been significantly improved in this revision, and all of this reviewer's comments have been addressed.
Author Response
Dear Reviewer,
Thank you so much for the comments on our paper.
This manuscript is a resubmission of an earlier submission. The following is a list of the peer review reports and author responses from that submission.
Round 1
Reviewer 1 Report
This study performed ion concentration analysis on a small number of bone samples retrieved from patients undergoing joint replacement surgery for osteoarthritis (OA). There are concerns regarding this study from the small amount of data collected, and only a single type of analysis was performed. The value of the study in informing the field is limited. Also, this study may not be suitable for publication in this journal as it is not relevant to environmental science or public health. Detailed comments are included below.
- Section 2.1 (methods) needs to be re-written to remove information that is describing the results, and include information on the procedures performed. E.g. “A history of disease did not affect the outcome of the study”; “The study included 22 patients with mild grade OA…The mean age and number of women and men did not differ…Significantly greater pain intensity was demonstrated in patients with higher radiological stage…” should all be moved to the results instead. This section should describe what patient and disease characteristics were collected, what inclusion and exclusion criteria were applied to determine which patients to include in the study, and how samples were collected and processed.
- Section 2.2 (methods) – although a study was cited, this section should still briefly describe how the bone samples were processed before analysis.
- The results only presented the concentrations of several elements from the bone samples analysed using ICP-OES. Very little analysis has been performed and only one type of analysis by itself is insufficient to demonstrate any kind of causal relationship between the concentration of a particular ion and the progression of OA disease. Although some speculations have been offered in the discussion, it is impossible to conclude from the data whether the changes in ion concentration are a cause or consequence of OA. Ion concentration in bone could be affected by a myriad of factors including the patient’s genetics, diet and lifestyle, and other possible conditions (particularly metabolic conditions). Given the important limitations of the study including the small sample size and lack of a control group, even the significant differences seen in the ion concentration analysis cannot be used to draw valid conclusions. It should also be noted that there is significant variation among the results within the sample group, as seen in the large standard deviations seen in the data, which further limits the ability to draw conclusions.
- The femoral head bone is only one component of the tissue that is affected in OA. Without detailed procedures for the extraction of bone tissue, it is unclear which part of the bone tissue was used for analysis (only the subchondral region, or the greater femoral head region). Theoretically the ion concentration in the subchondral bone region is the most relevant in OA pathogenesis and the rest of the femoral head is less relevant. Also, the ion concentration in other joint structures that are major players in OA, particularly the cartilage and synovial fluid, would have been highly relevant to analyse but this was not performed.
Reviewer 2 Report
Interesting analysis.
I have only one comment. Which are the data in healthy patients? Are the results really different from mild?
Reviewer 3 Report
The study by Dąbrowski et al. analyzes the content of trace elements in the head and the femoral neck of subjects suffering from osteoarthritis. English is very poor and not fluent, so it is difficult to fully follow the study. Patients: What do the authors mean when they write "A history of disease did not affect the outcome of the study."? What disease? The clinical and anthropometric characteristics of the patients (age, BMI, smoking habits, other diseases, drugs) are not reported. Results: the data presented in the text do not correspond to what is reported in table 1 (sometimes there is a confusion between median and mean). In the first row of Table 1, the headers should be "Femoral head" and "Femoral neck". Correlation data for patients with grade KL2 osteoarthritis are not reported in Table 2. Figure 1 is unclear. In the discussion, the authors suddenly write about patients suffering from RA (rheumatoid arthritis?), which obviously is a very different pathology from osteoarthritis Some acronyms (ZIP8, ColX) are not explicit. Sometimes the authors write "this study" but they mean other articles, not their study.